# The Influence of Insecure Attachment to Parents on Adolescents’ Suicidality

**DOI:** 10.3390/ijerph20042827

**Published:** 2023-02-05

**Authors:** Lara Leben Novak, Vanja Gomboc, Vita Poštuvan, Diego De Leo, Žiga Rosenstein, Maja Drobnič Radobuljac

**Affiliations:** 1Faculty of Medicine, University of Ljubljana, 1000 Ljubljana, Slovenia; 2Andrej Marušič Institute, Slovene Centre for Suicide Research, University of Primorska, 6000 Koper, Slovenia; 3Faculty of Mathematics, Natural Sciences and Information Technologies, Department of Psychology, University of Primorska, 6000 Koper, Slovenia; 4Unit for Intensive Child and Adolescent Psychiatry, Center for Mental Health, University Psychiatric Clinic Ljubljana, 1000 Ljubljana, Slovenia

**Keywords:** suicidal behavior, adolescents, attachment, parents, acquired capability for suicide

## Abstract

Insecure attachment has been identified as a risk factor for adolescent psychopathology and, consequently, for suicidal behavior. We aimed to highlight the relationship between the attachment styles of adolescents and their suicidal behavior and to investigate the role of each parent in the suicidality pathway of adolescents. The sample consisted of 217 adolescent inpatients who were at the highest risk for suicidal behavior and who were hospitalized in the Unit for Intensive Child and Adolescent Psychiatry. Self-report questionnaires assessing their attachment to their parents, their acquired capability for attempting suicide, their suicidality, and a number of traumatic life events were administered. The results showed a higher level of attachment avoidance rather than attachment anxiety among the most at-risk adolescents. An acquired capability for suicide (ACS) mediated the positive correlation between adolescents’ attachment avoidance in relation to the mother or father and their suicidality. The suppressive mediating effect of an ACS on the association between attachment anxiety in relation to the father and suicidality was detected. The odds ratio for attempted suicide was more than two times higher for adolescents who were insecurely attached to their father compared to adolescents who were insecurely attached to their mother. Our results confirmed the importance of attachment, especially paternal attachment, in developing suicidality during adolescence. Preventive and clinical interventions should target these important domains with the aim of decreasing suicidality among adolescents.

## 1. Introduction

Suicide is one of the leading causes of death among adolescents, which makes it an emerging global public health problem [1]. Slovenia is among the ten European countries with the highest suicide rate [2]. The reported suicide rate in those in the age range of 10 to 19 years was 2.14 (number of suicide deaths in a year per 100,000 population) [3].

Predicting suicidal behavior among teenagers is demanding because their suicidal risk is recognized as a multifaceted construct associated with a combination of protective factors and risk factors that are related to their neurobiological, social, psychological, and developmental resources [4,5]. New findings suggest that early life circumstances contribute to the prediction of suicidal behavior in adolescents. Some of the most exposed risk factors for suicidal behavior are the socioeconomic and demographic characteristics of the family, the parents’ psychological state, and parenting practices [6]. Other described risk factors are adolescents’ mental health (especially clinically significant depression, personality disorders, substance abuse, anxiety, etc.), the experience of adverse life events, negative peer dynamics, physical and psychological violence, sexual abuse, and a history of suicide attempts [7,8,9]. Various personality traits might also be important for determining suicidal behavior in adolescents. Common risk traits are neuroticism, introversion, irritability, anger, resentment, and low self-esteem [10]. Additionally, McCallum and her colleagues reported in a recent study the importance of four domains of suicidal behavior in adolescents—hopelessness, impulsivity, sensation seeking, and anxiety sensitivity [8]. Another very important factor for suicidality is also family functioning [8,11].

Thus, one of the explanations for the etiology of suicidal behavior is the attachment between adolescents and their parents. Attachment is an emotional bond between an individual and their attachment figure(s). Bowlby’s attachment theory states that early experiences with caretakers consolidate into internal representations of relationships (so-called internal working models) [12]. These serve as a guide for the interpretation of important relationships across the lifespan of a person and determine the person’s most important interactions [13,14]. Bowlby’s theory was expanded by Mary S. Ainsworth. Based on her study’s results, Ainsworth described three main attachment styles: one was described as secure, and two were described as insecure (anxious/ambivalent and avoidant) [15]. Later, the researchers Main and Solomon added to the first three organized styles a third disorganized insecure attachment style classification [16]. Secure attachment refers to the ability to trust and be trusted by others, and it gives one a sense of self-worth. Avoidantly attached adolescents develop a negative view of others due to unresponsive, rejecting, or intrusive and highly demanding parents, which makes them afraid of disappointment and which is why they prefer to remain distant and independent in close relationships [17]. They develop the belief that they need to be highly competent or nearly flawless at life tasks in order to maintain self-reliance and to avoid potential further rejection [13,18].

Adolescents with anxious/ambivalent attachment tend to seek closeness while, on the other hand, fearing abandonment [19]. Their parents are mostly inconsistent in responding to their needs and are highly critical of them. These adolescents tend to develop a negative internal model of the self and self-criticism, and they desperately need to obtain assurance and validation from others. Individuals with disorganized attachment behave contradictorily, indecisively, and inconsistently, as they were often exposed to adverse experiences in their childhood [15,19].

A caring and secure interpersonal relationship with one’s parents is one of the most important components in the maturation of a young person, the formation of attachment, and the development of the ability to regulate one’s emotions. If the parents’ responses are not in tune with the child’s needs, the child may develop less appropriate self-calming methods to cope with interpersonal challenges, including self-destructive behavior and aggressive acting-out behavior [20,21]. However, not all insecurely attached youth develop self-harm behavior [22].

In one of their studies, Cummins and her colleagues examined sensitivity to a range of sensory stimuli and altered the pain perception among individuals who performed self-harm. Their results showed that pain hyposensitivity was associated with self-harm frequency in adolescents with self-harm behavior [23]. Other studies suggested an elevated pain threshold in those who engage in self-injuring behavior [24,25]. Similarly, an insecure attachment to one’s parents is usually accompanied by physically and psychologically provocative life events that, over time, habituate an individual to pain. Thus, the exposure to trauma itself increases the pain threshold and theoretically leads to an acquired capability for suicide [26]. Also, Roley-Roberts, in her study, stated that trauma exposure might be linked to suicide risk via an increase in pain tolerance (i.e., an acquired capability for suicide) and the development of posttraumatic stress disorder via psychological disruption, which lead to perceived burdensomeness and thwarted belongingness [26]. These findings are consistent with the interpersonal theory of suicide, which posits that the simultaneous presence of two factors is needed for suicidal behavior—the capability to engage in suicidal behavior and the wish to die [27,28]. According to the theory, an acquired capability for suicide (ACS) is established through habituation to pain and the fear of death [28]. It develops as a consequence of repetitive painful and difficult life events that reduce the sensation of pain and the fear of injury and death overtime [27]. On the other hand, the wish to die develops in the presence of perceived burdensomeness and thwarted belongingness. The former consists of self-hate and the feeling of being a burden to others [28], while the latter consists of self-reported loneliness and a perceived absence of reciprocally caring relationships.

Taking into consideration both theories—the attachment theory and the interpersonal theory of suicide—some prior research discovered a connection between insecure attachment and increased suicidality [11,18,21,29,30,31,32,33]. Venta et al. [34], however, did not find such a connection. Instead, they confirmed the association between insecure attachment and internalizing disorders, which suggested that the relationship between attachment and suicidal behavior could be mediated by other factors. Proceeding from there, others investigated whether affective lability, an external locus of control [35], interpersonal difficulties [36,37], loneliness [36], self-criticism, dependency [17], feelings of entrapment [38], and reflective functioning [39] mediated this association. A mediating role was confirmed for interpersonal problems [36,37], feelings of loneliness [36], self-criticism [17], and the perception of entrapment in the contribution of anxious attachment to suicidal behavior [38]. The studies outlined different psychological factors that could intervene in the relationship between attachment style and suicidal behavior.

The majority of previous research examined insecure attachment to both parents or only to the mother, taking into account that, traditionally, the mother is regarded as the primary caretaker. One of the studies that separately explored the association between insecure attachment to the mother and to the father and suicidal ideation was performed by Waraan and her colleagues [40]. They reported a statistically significant association between suicidal ideation and a higher level of attachment anxiety in relation to the mother and the father in a clinical sample of adolescents with depression. When attachments to both parents were included in the same multivariate model, only attachment anxiety to the mother was significantly associated with suicidal ideation [40]. The latter finding is consistent with other research, reporting that only the attachment to the mother, but not the father, accounted for a significant variation in the level of suicidal behavior in adolescents [21,41]. However, none of the studies could conclude that attachment anxiety in relation to the father was not associated with suicidal risk [40]. Furthermore, Sheftall et al. found that the only significant predictor of attempted suicide in adolescents was an insecure attachment to the father [11].

Considering the above information, we aimed to further verify the association between attachment and suicidal behavior in the most at-risk adolescents and to explore whether an ACS may bridge this gap. To our knowledge, this association has never been explored on such a sample. We hypothesized that an insecure adolescent–parent attachment would predict a higher risk for adolescents’ suicidal behavior. We expected the mediator of this association to be an ACS. Furthermore, we wanted to analyze the role of the attachment of adolescents to each of their parents separately in the development of suicidal behavior. We hypothesized that there were no differences in the influence of attachment to the mother or to the father on the risk for suicidal behavior.

## 2. Materials and Methods

### 2.1. Participants and Procedures

This cross-sectional study was conducted on a sample of patients who had the highest risk for suicidal behavior and were consequently hospitalized in the Unit for Intensive Child and Adolescent Psychiatry at the University Psychiatric Clinic Ljubljana, which is the only secure psychiatric unit for children and adolescents in Slovenia. The collected sample consisted of adolescents with serious suicidal behavior, which covered a range of suicide-related activities, including suicidal ideation, thoughts, and plans as well as suicide attempts. Participants were aged between 11 and 18 years. The main exclusion criteria were intellectual disability, active psychosis, acute effects of illegal psychoactive substances including alcohol, and refusal to participate in the study.

The current study formed part of the “Do I understand (you)?!?—New approaches to identifying and preventing suicidal behavior and other risk factors in the field of adolescent mental health” project [42]. Enrolment began in July 2019 and ended in January 2022.

The participants were randomly selected considering the exclusion criteria. Patients received a thorough explanation of the purpose of the study and were invited to participate. In addition, they were informed that their data would be kept confidential. Informed consent was obtained from all interested participants and also from their parents if the participants were under the age of 15. After giving consent, the participants were administered several self-report questionnaires and were offered further explanations from the researcher. From 220 enrolled participants, 3 were excluded due to not completing all of the administered questionnaires. All the data were entered by two independent researchers to ensure minimal error in the transcription of the data. The research was approved by the National Medical Ethics Committee of the Republic of Slovenia, No: 0120-172/2019/4.

### 2.2. Measures

The Experiences in Close Relationships—Relationship Structures questionnaire (ECR-RS) was used [43]. This is a self-report scale that assesses attachment styles with respect to four close relationships (mother, father, romantic partner, and best friend) by asking the same 9 questions for each attachment figure, and it is scored on a 7-point Likert scale from 1 (strongly disagree) to 7 (strongly agree). The questionnaire measures two dimensions of attachment: anxiety and avoidance. Higher scores indicate greater insecurity of attachment. In our study, we measured attachment related to mother and father. The general adolescent–parent attachment was calculated as the mean value for both parents. The questionnaire had been validated for adolescents, but not specifically for Slovene adolescents [44]. However, it had been previously translated and already used in the Slovene adult population [45]. Total subscales had excellent internal consistency (Table 1).

The Paykel Suicide Scale (PSS) [46] was also used. It consists of four items that evaluate the presence of passive suicidal ideation, suicidal thought, and seriously considered suicidal plan and was scored from 0 (never) to 5 (always). The time frame refers to the last two weeks. Additionally, a question about suicide attempts was added. Higher scores indicate greater severity. The scale had been validated on samples of adolescents, but not Slovene adolescents [47,48]. It had excellent internal consistency (Table 1).

A short version of the Acquired Capability for Suicide Scale (ACSS) [49] was used. It contains 7 items related to aspects of acquired ability for attempting suicide and to an individual’s pain tolerance and fear of death. All items are rated on a scale of 0 to 4, with higher scores indicating greater levels of fearlessness about death. The scale had good internal consistency (Table 1). The scale had been validated on a foreign sample of adolescents at high risk for suicidal behavior [50].

Adolescents also completed the Lifetime Incidence of Traumatic Events—Self-Report scale (LITE-S) [51]. This 16-item questionnaire asks about a broad range of traumatic events and events related to loss, such as physical or emotional abuse, accidents, bullying, and the loss of a close family member through divorce or death, as well as their emotional impact. The scale had good internal consistency in the present sample (Table 1). The scale had been previously validated on a sample of Slovene adolescents [52].

### 2.3. Data Analysis

The descriptive statistics of the sample were calculated using frequency and descriptive statistics. To assess whether the correlation between attachment style and suicidality was mediated by acquired capability for suicide, the mediation analyses were applied using Hayes’ PROCESS macro version 4 for SPSS model 4 [53]. Average scores of LITE-S questionnaires were included as covariates in the mediation model to adjust for the potential effects of adverse life experiences.

Assumption checks were conducted to assess for outliers, linearity, homoscedasticity, and nonnormal distribution of data. Five thousand bootstrapped samples were drawn to generate a sampling distribution, and a 95% confidence interval was reported for the indirect effect. Statistical significance of the indirect effect, calculated by using bootstrapped samples, was determined by the absence of zero from the confidence interval [54].

Statistical Package IBM SPSS Statistics for Windows version 28.0 [55] was used to analyze the data obtained in the study. A significance level of *p* < 0.05 was used.

A chi-square test of independence was used to compare the risk of suicide attempt between adolescents who were securely and insecurely attached to mother and to father. The odds and the odds ratio were calculated.

## 3. Results

### 3.1. Descriptive Analysis

The study included 217 adolescents aged 11 to 18 years with a mean age of 15.8 ± 1.5 years. Of these, 178 (82.0%) were female, and 39 (18.0%) were male. Less than a fifth of the patients were generally securely attached. The reason for hospitalization was experiencing uncontrollable suicidal thoughts for 6.9% of the adolescents; more than half of the sample (55.8%) also had a suicide plan, and a third (37.3%) had attempted suicide. Two thirds (64.5%) of patients reported a lifetime experience of attempting suicide. The most frequent discharge diagnosis was a developing personality disorder followed by depression, anxiety disorders, adaptive disorder, behavioral and emotional disorders in childhood and adolescence, and eating disorders (Table 2). Descriptive statistics for all questionnaire measures are reported in Table 3.

### 3.2. The Effect of Adolescent Attachment to the Mother and the Mediating Effect of the Adolescent Acquired Capability for Suicide

The scores of the ECR-RS avoidance subscale were positively correlated with suicidal behavior (*b* = 0.53, *t* (N = 176) = 2.3, and *p* = 0.022). Also, higher attachment avoidance predicted a greater adolescent ACS, and a greater ACS predicted suicidal behavior. When the adolescent ACS was entered as the mediator, a significant coefficient was revealed. Therefore, an indirect effect of attachment avoidance in relation to the mother on suicidal behavior via an ACS was found to be statistically significant. Because the direct and indirect effects were significant and because the coefficient of the direct effect model decreased but still existed, we could conclude that an ACS partially mediated the relationship between adolescent attachment avoidance in relation to the mother and suicidal behavior. The association between attachment avoidance in relation to the mother and adolescent suicidal behavior with the mediating effect of an ACS is demonstrated in Figure 1.

The process was repeated with the use of the scores of the ECR-RS anxiety subscale of attachment to the mother (Table 4). As attachment anxiety was not significantly associated with an ACS, there was no ground for mediation, so it was not explored further as a mediating variable.

Self-reported adverse life experiences measured with the LITE-S questionnaire were included as a covariant in all the mediation models to adjust for their potential effect on the association between attachment to the mother and suicidality. No significant effects were found.

### 3.3. The Effect of Adolescent Attachment to the Father and the Mediating Effect of the Adolescent Acquired Capability for Suicide

Similarly, an adolescent ACS statistically significantly mediated the effect of attachment avoidance in relation to the father on adolescent suicidal behavior. Both variables had significant and direct positive effects on suicidality as well. There was also a significant positive effect of attachment avoidance on the ACS (Figure 2).

There was a negative but significant correlation between attachment anxiety in relation to the father and an ACS. Namely, the higher the attachment anxiety, the lower the ACS. However, the isolated effects of attachment anxiety and an ACS on suicidal behavior were both statistically significantly positive. The indirect effect of adolescent attachment anxiety in relation to the father on suicidal behavior via an ACS was found to be statistically significantly negative, suggesting a suppressive relationship, with the mediator (i.e., ACS) being the suppressor (Figure 3).

Self-reported adverse life experiences measured with the LITE-S questionnaire were included as a covariant in all the mediation models to adjust for their potential effect on the association between attachment to the father and suicidality. No significant effects were found.

### 3.4. The Role of the Security of Attachment to the Parents in Adolescent Attempted Suicide

In a categorical assessment, only the security of attachment to the father was significantly correlated with adolescent attempted suicide (χ^2^ (1) = 5.85, and *p* = 0.02). The security of attachment to the mother (χ^2^ (1) = 0.01, and *p* = 0.92) and the combined security of attachment to both parents (χ^2^ (1) = 1.36, and *p* = 0.24) were not significantly correlated (Table 5 and Table 6). The odds for a lifetime experience of attempting suicide in the group of adolescents insecurely attached to the father were higher than the odds for those who were insecurely attached to the mother (Table 7). The odds ratio for a lifetime experience of attempting suicide were almost three times higher for individuals who were insecurely attached to the father compared to individuals who were insecurely attached to the mother (Table 7).

## 4. Discussion

The present study aimed at expanding the knowledge on the relationship between adolescent attachment to the parents and suicidality in the most at-risk adolescents. Another aim was to assess the possible mediating role of the acquired capability for suicide in this relationship.

Our results showed that insecure attachment to the father especially increased the odds of adolescent attempted suicide. They also showed that adolescent attachment avoidance, not anxiety, in relation to each of the parents correlated with adolescent suicidality, influenced the ACS, and directly and indirectly increased the probability of suicidal behavior.

The results of our study, which confirmed the relationship between insecure attachment to the parents and adolescent suicidal behavior, were consistent with the findings from previous research. The results, which supported the expected mediation effect of an ACS in the case of attachment avoidance, were also consistent with prior evidence showing that attachment avoidance, but not anxiety, predicted the suicide attempt status among adolescents after controlling for family alliance and depressive symptoms [11]. A similar relationship was found among a sample of adults, where Grunebaum et al. [30] reported that attachment avoidance predicted a greater risk of a suicide attempt after presentation with a major depressive episode. The results could be explained by the fact that each pattern of attachment results in unique strategies for coping with stressful life circumstances [12,15], which leads to differences in expressing distress and managing life’s challenges. Adolescents with higher attachment avoidance may have experienced more painful physical or psychological adverse events in their relationships with their insecurely attached parents and, consequently, may habituate to the fear and pain over time, which increases the risk for an acquired capability for suicide [56]. When they encounter interpersonal difficulties or fail to reach their own standards, they might not ask others for help but instead react with an inappropriate reaction, such as suicidal behavior [11,13,57]. Our results, which showed a positive association between attachment avoidance in relation to both parents and an ACS, were in line with these considerations.

On the contrary, adolescents with an expressed attachment anxiety dimension view others as able to provide support if their attention can be secured and retained. The threshold at which one’s level of fear and pain insensitivity manifests itself in one’s capability for suicide might not be raised and does not contribute to suicidal risk as much as in the case of avoidant attachment [56]. Therefore, it was suggested that other factors may be more important in mediating the association between attachment anxiety and suicidal behavior. Taking into consideration the results of other research [11,58], where attachment anxiety highly correlated with depressive symptoms while their correlation with attachment avoidance was only modest, depressive symptoms might be a mediator of this association. This warrants further evaluation.

Moreover, the present study surprisingly revealed an indirect suppression effect of attachment anxiety in relation to the father on adolescent suicidal behavior which occurred by decreasing an adolescent’s acquired capability for suicide. However, despite the suppressive indirect effect of attachment anxiety on suicidal behavior, the results showed a significant and direct positive correlation between attachment anxiety in relation to the father and adolescent suicidal behavior, and, therefore, it still represents a significant risk factor for suicidality. The reason for such results might be that the association is likely to be mediated by other constructs which dominate the ACS according to the findings of Kacmarski et al. [59] who found attachment anxiety to be significantly related to perceived burdensomeness and thwarted belongingness but not to an ACS. These more interpersonally focused constructs of the interpersonal theory of suicide might be the mediators of the effect of attachment insecurity on suicidal behavior.

It is of interest that, in our study, higher attachment avoidance was revealed as more important in relation to suicidal behavior, whereas many studies have found suicidality to be associated with attachment anxiety rather than avoidance [41,60,61]. In our group of the most at-risk adolescents, the avoidance dimension was higher than the anxiety dimension of attachment. Taking into consideration the results of previous research, which showed a connection between attachment avoidance and attempted suicide but not suicidal ideation [11,30], and according to the fact that our study was performed on a sample of adolescents who were hospitalized in a secure psychiatric unit, it seems that attachment avoidance may be a more specific marker for adolescents with a higher risk for suicide. If so, it could be an important target in the field of suicide prevention.

Furthermore, a statistical analysis revealed the correlation between adolescent attachment and attempted suicide only in the case of attachment to the father but not to the mother. Thus, the risk for attempted suicide was significantly higher among insecurely attached adolescents in relation to the father as compared to adolescents who were insecurely attached to the mother, and this showed the importance of the attachment to the father in the light of attempting suicide. According to these results, we might conclude that each parent provides a different type of emotional support and, consequently, has a different impact on adolescent suicidal behavior.

When interpreting these results, it is important to consider that the questionnaires were not given to all the patients at the beginning of their hospitalization; thus, their responses might have been influenced by the improvement and stabilization of their mental state. Secondly, all questionnaires were self-reported and did not represent the most objective point of view. Here, we can encounter the common limitation of self-report questionnaires, which is called social desirability bias [62], especially because adolescents usually have a tendency to answer questions in a manner that will be viewed favorably by the clinician. Thirdly, except for the LITE-S questionnaire [51,52], the remaining scales were not validated specifically on a sample of Slovene adolescents. However, they had been previously officially translated to the Slovenian language and showed good to excellent internal consistency with other languages [44,47,48,50]. Also, the LITE-S [51] and ACSS [49] questionnaires did not have high Cronbach’s alpha internal consistencies, but their values were still within the acceptable level. The results should be interpreted with caution. Another limitation needing to be addressed is that a more accurate family structure was not evaluated in the analysis since more than 60% of the patients were living with both their biological parents and since an additional 38% were living with one of their parents with or without a stepparent. Therefore, the generalizability of the study findings to different family structures is limited. Also, the sample is derived from a clinical population with the highest risk for suicidal behavior and, therefore, cannot be considered to be representative of the general population. This study was cross-sectional and could not support causal inferences. A longitudinal study would provide a better understanding of the influences on adolescent suicidal behavior.

However, the results of our study provided important new information on the association between the adolescent attachment to the parents and suicidal behavior that could improve the recognition of suicide risks and help-seeking behavior through the recognition of the causes and risk factors of suicidal behavior. The dimensions of attachment avoidance and anxiety show different mechanisms leading to suicidal behavior in the most at-risk adolescent inpatients. The determination of the attachment dimension may serve as a screening method to predict the level of suicide risk and to provide important prognostic information about later suicidal behavior [18]. Furthermore, more specific interventional trials could be provided if the attachment dimension was assessed. Thus, families would get more specific directions to meet the needs of their children. One of the empirically supported approaches aiming to repair the adolescent–parent attachment and to therewith decrease adolescent suicidality is attachment-based family therapy (ABFT). This therapy helps to resolve ruptures in family communication, improves the supportive and organizational function of the family, teaches parents how to refine effective caregiving behaviors, and helps adolescents to increase their capacity for problem solving and affect regulation [63,64,65].

While other interventions aimed at increasing adolescents’ attachment security may prove protective in future studies [39,66], early interventions protecting children from exposure to painful life events and, therefore, from habituation to fear and pain (thus acquiring the capability for suicide) might be crucial [56], especially for those with a higher dimension of attachment avoidance.

## 5. Conclusions

Our study confirmed that adolescent attachment may operate as a general vulnerability factor, increasing the risk for suicidal behavior [67]. Attachment avoidance, especially in relation to the father, was revealed as an important factor in predicting suicidal behavior among the most at-risk adolescents. An acquired capability for suicide was confirmed to be the mediator of the association between avoidance in the adolescent–parent attachment and suicidal behavior. According to the results, providing support to parents in the caregiving process would be a significant preventive approach. However, due to the complexity of suicidal behavior in adolescents, one intervention strategy by itself may not be sufficient. Also, encouraging and increasing the involvement of fathers in parenting demands greater attention [68]. This also underscores the need for more research exploring the attachment to the father and its association with suicidal behavior [40]. In clinical settings, the identification of the attachment style may inform clinicians in better understanding adolescents’ needs and in establishing appropriate interventions to protect adolescents from suicidal behavior [30,40,60,69].

## Figures and Tables

**Figure 1 ijerph-20-02827-f001:**
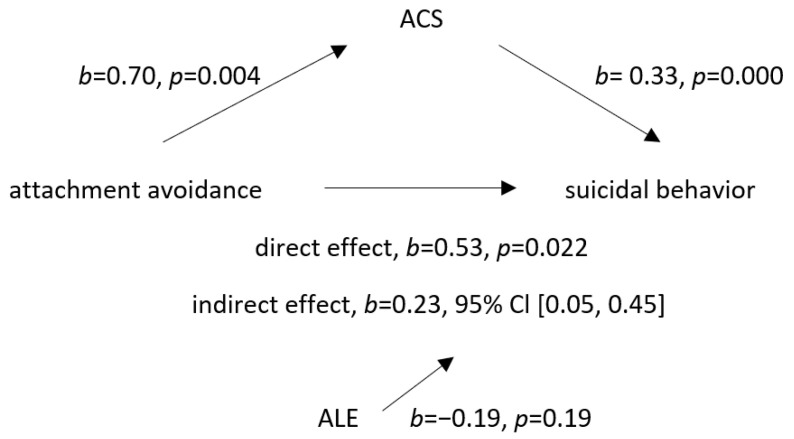
Mediation model with acquired capability for suicide (ACS) as the mediator of the association between attachment avoidance to mother and suicidal behavior. Adverse life experiences were included as a covariant. Legend: ACS—acquired capability for suicide, ALE—adverse life experiences, direct effect—association between attachment avoidance and suicidal behavior, and indirect effect—association between attachment avoidance and suicidal behavior when ACS was added as the mediating variable.

**Figure 2 ijerph-20-02827-f002:**
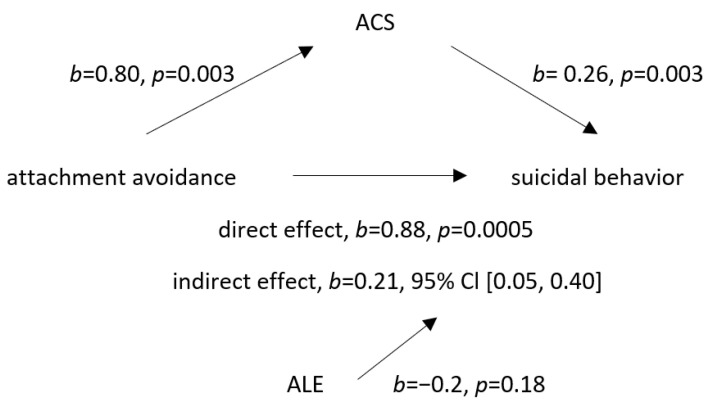
Mediation model with acquired capability for suicide (ACS) as the mediator of the association between attachment avoidance in relation to father and suicidal behavior. Adverse life experiences were included as a covariant. Legend: ACS—acquired capability for suicide, ALE—adverse life experience, direct effect—association between attachment avoidance and suicidal behavior, and indirect effect—association between attachment avoidance and suicidal behavior when ACS was added as the mediating variable.

**Figure 3 ijerph-20-02827-f003:**
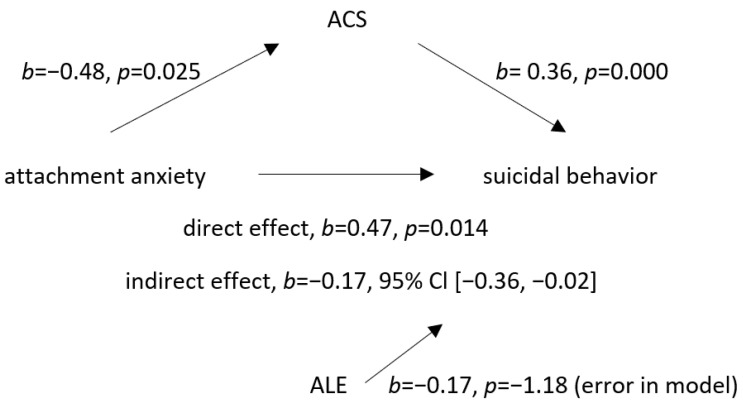
Mediation model with acquired capability for suicide (ACS) as the mediator of the association between attachment anxiety in relation to father and suicidal behavior. Adverse life experiences were included as a covariant. Higher attachment anxiety in relation to father decreased ACS and, consequently, decreased suicidal behavior, while attachment anxiety in relation to father directly increased suicidal behavior. Legend: ACS—acquired capability for suicide, ALE—adverse life experience, direct effect—association between attachment anxiety and suicidal behavior, and indirect effect—association between attachment anxiety and suicidal behavior when ACS was added as the mediating variable.

**Table 1 ijerph-20-02827-t001:** Internal consistency of used questionnaires.

	α
ECR-RS	0.82–0.88
PSS	0.87
ACSS	0.67
LITE-S	0.60

Legend: α refers to Cronbach’s alpha coefficient, ECR-RS refers to Experiences in Close Relationships—Relationship Structures questionnaire, PSS refers to Paykel Suicide Scale, ACSS refers to Acquired Capability for Suicide Scale, and LITE-S refers to Lifetime Incidence of Traumatic Events—Self-Report Scale.

**Table 2 ijerph-20-02827-t002:** Characteristics of the patients.

	Number of Participants	Proportion
diagnosis		
developing personality disorder (PD) *	99	45.6%
depressive mood disorder (DMD)	86	39.6%
PD and DMD	31	14.3%
anxiety disorder	26	12.0%
adaptive disorder	26	12.0%
behavioral and emotional disorder	25	11.5%
non-suicidal self injuring behavior		
anytime in life	184	84.8%
in the last 3 months	159	73.3%
hospitalization		
past hospitalization	62	28.6%
attachment		
secure	33	17.9%
insecure	151	82.1%
adverse life experience		
parental divorce	86	39.6%
violence among parents	49	22.6%
experience of being abused	98	45.2%
adults living within the same household		
living with both parents	133	61.3%
living with mother and/without stepfather	62	28.6%
living with father and/without stepmother	20	9.2%
living with grandparents only	2	0.9%
education level		
primary school	78	35.9%
secondary school	139	64.1%
ethnicity		
Slovene	207	95.4%
Balkan ethnic groups	6	2.8%
others	4	1.8%

* Prefix “developing” in the diagnosis of PD was added, suggesting that the personality disorder was not present in its complete dimensional conceptualization of the disorder but that some specific personality features of functioning and behavior had already been present since the diagnoses were made according to the ICD-10 criteria.

**Table 3 ijerph-20-02827-t003:** Descriptive statistics of the sample.

	Min	Max	Mean	SD
duration of hospitalization (days)	2	23	6.7	4.2
ECR-RS anxiety dimension in relation to mother	1	7	3.4	2.0
ECR-RS avoidance dimension in relation to mother	1	7	4.0	1.5
ECR-RS anxiety dimension in relation to father	1	7	3.1	1.9
ECR-RS avoidance dimension in relation to father	1	7	4.8	1.4
ECR-RS anxiety dimension in relation to parents	1	7	3.2	1.8
ECR-RS avoidance dimension in relation to parents	1	7	4.4	1.2
LITE-S	0	11	4.1	2.5
ACSS	4	28	18.1	5.1
PSS	0	20	13.3	5.3

Legend: min refers to minimum score, max refers to maximum score, SD refers to standard deviation, ECR-RS refers to Experiences in Close Relationships—Relationship Structures questionnaire, LITE-S refers to Lifetime Incidence of Traumatic Events—Self-Report Scale, ACSS refers to Acquired Capability for Suicide Scale, and PSS refers to Paykel Suicide Scale.

**Table 4 ijerph-20-02827-t004:** Linear regression between attachment anxiety in relation to mother and acquired capability for suicide.

Way	*b*	SE	*t*	*p*	LCl	UCl
constant	20.15	0.94	21.39	0.000	18.29	22.01
ECR-RS anxiety subscale	−0.18	0.19	−0.96	0.34	−0.56	0.19
LITE-S	−0.32	0.15	−2.08	0.04	−0.63	−0.02

Legend: SE refers to standard error, LCl refers to lower control limit, UCl refers to upper control limit, ECR-RS refers to Experiences in Close Relationships—Relationship Structures questionnaire, LITE-S refers to Lifetime Incidence of Traumatic Events—Self-Report Scale.

**Table 5 ijerph-20-02827-t005:** Patients admitted to the psychiatric secure unit according to security of attachment to mother and lifetime experience of attempting suicide. Data are number of patients (%).

	Attachment	Secure	Insecure	
Attempted Suicide				
yes		30 (65.2)	96 (64.4)	126 (64.6)
no		16 (34.8)	53 (35.6)	69 (35.4)
		46 (23.6)	149 (76.4)	195 (100%)

**Table 6 ijerph-20-02827-t006:** Patients admitted to the psychiatric secure unit according to security of attachment to father and lifetime experience of attempting suicide. Data are number of patients (%).

	Attachment	Secure	Insecure	
Attempted Suicide				
yes		11 (44.0)	112 (68.7)	123 (65.4)
no		14 (56.0)	51 (31.3)	65 (34.6)
		25 (13.3)	163 (86.7)	188 (100%)

**Table 7 ijerph-20-02827-t007:** Odds for lifetime experience of attempting suicide according to security of attachment to mother and father and odds ratio for lifetime experience of attempting suicide.

	Parent	Mother	Father
Attachment			
secure		1.88	0.79
insecure		1.81	2.20
odds ratio		0.96	2.79

## Data Availability

The data that support the findings of this study are available from the corresponding author upon reasonable request.

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
