# Peer review of "The Influence of Insecure Attachment to Parents on Adolescents’ Suicidality"

_ijerph, 2023, doi:10.3390/ijerph20042827_

Round 1

Reviewer 1 Report

The title should better describe the objective of the study.

Improve the theoretical update. See, for example:

https://www.mdpi.com/1660-4601/16/22/4496

https://doi.org/10.5093/ejpalc2018a2

http://doi.org/10.11606/s1518-8787.2020054001637

https://doi.org/10.1016/j.jad.2022.05.010

https://doi.org/10.1016/j.jad.2022.05.029

Clinical instruments with an α = .60 and .67 is low. Justify why they are used.

More description of the procedure and sample is missing.

Review the journal's guidelines.

Expand the conclusions a little more. Also specify implications in the forensic field.

Author Response

Author´s Responses to the Reviewer

The authors sincerely thank the Reviewer for the valuable comments.

Comment 1: The title should better describe the objective of the study.

The title was changed to better represent the objective of the study: ˝The Influence of Insecure Attachment to Parents in the Suicidal Pathway of Adolescents˝.

Comment 2: Improve the theoretical update. See, for example.

More theoretical information from suggested sources was added in the introduction section (lines: 39-51, 122-135).

Comment 3: Clinical instruments with an α = .60 and .67 is low. Justify why they are used.

Internal consistency of Chronbach´s alpha is questionable, however, it is generally accepted, that α of 0.6-0.7 indicates an acceptable level of reliability. Therefore, these two clinical instruments/questionnaires were still used. This is additionally mentioned in the section of limitations of the study (lines: 406-408).

Comment 4: More description of the procedure and sample is missing. Review the journal's guidelines.

We provided some more information of the sample and more in detail description of the study´s procedure was added as suggested (lines 160, 163, 166-168).

Comment 5: Expand the conclusions a little more. Also specify implications in the forensic field.

The conclusion was expanded, and some amendments were made as suggested (lines: 439-449). But we are not sure on what ˝implications in the forensic field˝ refer to because the law of forensic science varies by country?  We would be grateful if the reviewer would be so kind to more specifically defined it.

Reviewer 2 Report

Brief summary: This is an interesting study that tackles the notion of the association between attachment (parent-child) and suicidal behavior in suicidal adolescents as well as exploring acquired capability for suicidal behavior (habituation to pain and fear of death). The authors sampled 217 adolescents’ inpatients with the highest risk for suicidality and they administered self-report questionnaires assessing attachment to parents, acquired capability for attempting suicide, suicidality, and traumatic life-events. They reported several interesting results (as listed below) and conclude that clinicians should target the domains of attachment and acquired capability for suicide to decrease suicidality among adolescents.

Overall, the authors portrayed a well-designed study and provided an integrative study that is pertinent to the field of suicidality in the youth.

Please see below my comments. Despite the numerous comments made below, this study is very well-written and is important to the readership.

Introduction:

Lines 33-34 : A reference is missing. It would be interesting for international readers to have an idea of the prevalence of suicide in Slovenia rather than specifying ‘’highest suicide rate’’ as the readership might not be familiar with Slovenia’s situation.

Lines 32-36: Considering that suicide is the main them of this study, it is encouraged that the authors develop the notions known on suicidality among adolescents. Paragraph 3 and following of the introduction focus rapidly on attachment and interactions aetiologies. However, the common risk factors of suicide should be explicated for the readership.

Lines 36-39 : In the last decade, predictive medicine has attempted to solve this question by designing prediction models.  For example, see Navarro’s study on the subject: https://jamanetwork.com/journals/jamanetworkopen/fullarticle/2777426  . This could provide additional information for the readers as to ‘’why it is needed to further develop our understanding of suicidality amongst adolescents’’.

Line 41-43: Attachment is defined differently across different significant figures in the field of psychology. From your references, it is understood that you are referring to Bowlby’s definition of attachment (line 42). It is encouraged to explicitly state it as you are referring later to Ainsworth’s work which is building upon this.

Line 58-59: Missing a reference.

Lines 93-95: The introduction of this paragraph is misleading the readers in believing that what is coming up is a literature review, which is not the case. It is suggested to rephrase the introductory sentence considering it is not needed to announce what you plan to do in the upcoming sentences.

Last paragraph: It would be interesting for the authors to state their hypothesis about their aim as this could be part of the discussion as well later on in the manuscript, considering their expertise in the field.

Material and methods:

Paragraph 1 : WHO defined adolescents to be between ages of 10 and 19 years old (La limite d’âge entre l’adolescence et l’âge adulte. Paediatr Child Health. 2003;8(9):578.). Why were adolescents in the presented study between 11 and 18? It is suggested, even if it appears trivial, to state the definition used to refer to an adolescent as this study focuses on these individuals.

Questionnaires: It is suggested to account for clarity that the metrics presented for the individual questionnaires are put into a table form to express validity and reliability instead of a continuous text. This way, the authors can solely define the questionnaires and explain their purpose while the metrics can all reside in a single area of the manuscript.

Data collection: It is understood that these are paper questionnaires that are reported to SPSS for the statistical analysis. How is this task conducted and who is responsible for ensuring data integrity (i.e: ensure that there is no error in transcription that might affect the results, etc).

Otherwise, the Material & Methods section is very clear and logical as per the aim the authors are attempting to achieve.

Results:

Patients` characteristics: Important variables are missing in Table 1 such as the age of the participants (mean, min, max) considering on a psychodynamic level the understand of different attachment and relationships dynamics differ between someone who is 11 years old and someone who is 18. This will help the readers in understanding ‘’who are’’ the participants. Number of years of education is also relevant here considering the dynamics with authority figures.

Line 209: a bracket is missing.

Minor comment : Table 4 & 5 – the title of the first column is crossed by a black bar which makes it confusing for the readership.

Otherwise, the result section is very clear.

Discussion:

The discussion is very relevant and uses appropriate literature to support the authors argument.

Limitations are stated between lines 352 and 367. A common limitation with self-reported questionnaire is the recall-bias and the difficulty related to ‘’attempting to please to the clinician’’ that is often found with certain personality traits. These could be added to the discussion.

Conclusion:

The conclusion could be bonified by restating explicitly the main results of your study as they are very relevant and add to the reasoning behind the last sentence of your paragraph.

Minor comment: There is a divergence in the definition of ACS across the manuscript and the abstract. In the manuscript ACS refers to Acquired Capability for Suicidal behavior whereas in the abstract it refers to Acquired Ability for Suicide (AAS?). This should be addressed.

Author Response

Author´s Responses to the Reviewer:

The authors sincerely thank the Reviewer for the valuable comments.

Comment 1: Lines 33-34 : A reference is missing. It would be interesting for international readers to have an idea of the prevalence of suicide in Slovenia rather than specifying ‘’highest suicide rate’’ as the readership might not be familiar with Slovenia’s situation.

The reference was added. The suicide rate in Slovenia has already been reported for the ages ranged from 10 to 19 years, which was 2.14 deaths per 100 000.

Comment 2: Lines 32-36: Considering that suicide is the main them of this study, it is encouraged that the authors develop the notions known on suicidality among adolescents. Paragraph 3 and following of the introduction focus rapidly on attachment and interactions aetiologies. However, the common risk factors of suicide should be explicated for the readership.

Taking your comment into consideration, the common risk factors for suicidality among adolescents were added in the introduction (lines: 39-51).

Comment 3: Lines 36-39 : In the last decade, predictive medicine has attempted to solve this question by designing prediction models.  For example, see Navarro’s study on the subject: https://jamanetwork.com/journals/jamanetworkopen/fullarticle/2777426  . This could provide additional information for the readers as to ‘’why it is needed to further develop our understanding of suicidality amongst adolescents’’.

More information about risk factors for suicidal behavior among adolescents was added (lines: 39-51).

Comment 4: Line 41-43: Attachment is defined differently across different significant figures in the field of psychology. From your references, it is understood that you are referring to Bowlby’s definition of attachment (line 42). It is encouraged to explicitly state it as you are referring later to Ainsworth’s work which is building upon this.

We more explicitly stated the contributions of different significant professionals in a field of attachment as was recommended (lines: 59-65).

Comment 5: Line 58-59: Missing a reference.

The reference was added.

Comment 6: Lines 93-95: The introduction of this paragraph is misleading the readers in believing that what is coming up is a literature review, which is not the case. It is suggested to rephrase the introductory sentence considering it is not needed to announce what you plan to do in the upcoming sentences.

The introductory sentences of the mentioned paragraph were rephrased (lines: 109-110).

Comment 7: Last paragraph: It would be interesting for the authors to state their hypothesis about their aim as this could be part of the discussion as well later on in the manuscript, considering their expertise in the field.

The hypothesis about the aims of the study were written in the last paragraph of the introduction (lines 137-140, 142-144).

Comment 8: Paragraph 1 : WHO defined adolescents to be between ages of 10 and 19 years old (La limite d’âge entre l’adolescence et l’âge adulte. Paediatr Child Health. 2003;8(9):578.). Why were adolescents in the presented study between 11 and 18? It is suggested, even if it appears trivial, to state the definition used to refer to an adolescent as this study focuses on these individuals.

The sample consist of patients between the ages of 11 and 18 years old. The lower limit of 11 years of age was set as a result of the youngest adolescents who have been hospitalized in the Unit for Intensive Child and Adolescent Psychiatry due to the suicidal behavior/risk. The upper limit was set considering the Unit does not accept patients older than 18 years of age. Such patients are hospitalized in the adult psychiatry units.

Comment 9: Questionnaires: It is suggested to account for clarity that the metrics presented for the individual questionnaires are put into a table form to express validity and reliability instead of a continuous text. This way, the authors can solely define the questionnaires and explain their purpose while the metrics can all reside in a single area of the manuscript.

New table was designed containing values of internal consistency of the questionnaires, as suggested (Table 1).

Comment 10: Data collection: It is understood that these are paper questionnaires that are reported to SPSS for the statistical analysis. How is this task conducted and who is responsible for ensuring data integrity (i.e: ensure that there is no error in transcription that might affect the results, etc).

The additional data of the procedure was described (lines 166-168).

Comment 11: Patients` characteristics: Important variables are missing in Table 1 such as the age of the participants (mean, min, max) considering on a psychodynamic level the understand of different attachment and relationships dynamics differ between someone who is 11 years old and someone who is 18. This will help the readers in understanding ‘’who are’’ the participants. Number of years of education is also relevant here considering the dynamics with authority figures.

The age of the participants is represented in the written paragraph of the results section, therefore, we did not want to repeat the information in the Table 1 or 2. We do not have data for the number of educational years, although information of the education level is provided in Table 1.

Comment 12: Minor comment: Table 4 & 5 – the title of the first column is crossed by a black bar which makes it confusing for the readership.

Tables have been graphically corrected.

Comment 13: Limitations are stated between lines 352 and 367. A common limitation with self-reported questionnaire is the recall-bias and the difficulty related to ‘’attempting to please to the clinician’’ that is often found with certain personality traits. These could be added to the discussion.

Suggested limitation in the discussion section was added (lines 400-402).

Comment 14: The conclusion could be bonified by restating explicitly the main results of your study as they are very relevant and add to the reasoning behind the last sentence of your paragraph.

We expanded the conclusion restating the main results of the study, as suggested (lines: 439-449).

Comment 15: Minor comment: There is a divergence in the definition of ACS across the manuscript and the abstract. In the manuscript ACS refers to Acquired Capability for Suicidal behavior whereas in the abstract it refers to Acquired Ability for Suicide (AAS?). This should be addressed.

The term "Acquired Capability for Suicide" was unified across the manuscript.

Reviewer 3 Report

The authors explore the relationship between attachment styles to their parents and suicidal behavior, as well as the mediating role of ACS. The results provide meaningful perspective for the understanding of adolescent suicidal behavior and contribute to the establishment of parent-child relationship. There are some key concern need to be cleared.

1. My first and primary concern lies in the novelty of this work, as I feel that the novelty issue has not been sufficiently highlighted in the current introduction. An important question shall be answered: does this work fill up some knowledge gaps which previous articles cannot address?

2. The research seemingly overlooked the literature which distinguishes the influence of adolescent attachment to “father” and “mother” on suicidal behavior. The authors may miss some detailed knowledge on what is known on the topic based on prior studies and what needs to be known.

3. The results showed that the odds ratio for attempted suicide was more than twice higher for adolescents insecurely attached to their father compared to adolescents insecurely attached to their mother, but the proportion of female (82.0%) and male (18.0%) subjects were different. Please expand the discussion.

Author Response

Author´s Responses to the Reviewer:

The authors sincerely thank the Reviewer for the valuable comments.

Comment 1:  My first and primary concern lies in the novelty of this work, as I feel that the novelty issue has not been sufficiently highlighted in the current introduction. An important question shall be answered: does this work fill up some knowledge gaps which previous articles cannot address?

To our knowledge, there have been no studies that would verify the association between attachment and suicidal behavior in the most-at-risk inpatient adolescents. The role of attachment of adolescents to each of their parents was analyzed separately in the developmental stage of suicidal behavior since not many studies exploring such connections are accessible. We highlighted that in the introduction (lines: 138-140).

Comment 2: The research seemingly overlooked the literature which distinguishes the influence of adolescent attachment to “father” and “mother” on suicidal behavior. The authors may miss some detailed knowledge on what is known on the topic based on prior studies and what needs to be known.

The overlook of the present literature distinguishing the influence of adolescent attachment separately to mother and father on suicidal behavior was added as suggested (lines: 122-135).

Comment 3: The results showed that the odds ratio for attempted suicide was more than twice higher for adolescents insecurely attached to their father compared to adolescents insecurely attached to their mother, but the proportion of female (82.0%) and male (18.0%) subjects were different. Please expand the discussion.

Mentioned proportion of female and male referred to adolescent participants. For each, the attachment to mother and father was specified, followed by calculations of odds for lifetime attempted suicide according to security of attachment to mother and father and odds ratio for lifetime attempted suicide.

Round 2

Reviewer 3 Report

I have no more comment and recommend the publication of this manuscript.